# Shift is Good: Mismatched Data Mixing Improves Test Performance

## Abstract

We consider training and testing on mixture distributions with different training and test proportions. We show that in many settings, and in some sense generically, distribution shift can be beneficial, and test performance can improve due to mismatched training proportions. In a variety of scenarios, we identify the optimal training proportions and the extent to which such distribution shift can be beneficial.

## 1 Introduction

Imagine that you are taking a high-stakes exam next week. The exam will be 90% on European history and 10% on Chinese history. Both topics are equally familiar to you and equally difficult, and additional study will help you with each topic similarly. You have unlimited access to study material and practice questions for both. How should you spend your limited studying budget? Should your training match your test distribution, studying 90% European and 10% Chinese? Or would you benefit from a distribution shift? Studying more Chinese history? Less? Only European history? *We encourage the reader to pause and make an intuitive guess.*

The answer depends on the specific learning curve for improvement in test performance within a topic as a function of the number of training examples from that topic. But at least for a generic $1/n$ scaling (as obtained from e.g., both learning VC classes and in parametric regression), the answer, as we will see in Section 3, is that you would benefit from a distribution shift, and should study 75% European History and 25% Chinese history—this would reduce your test error by 20% over the $90/10$ non-shifted training.

We just saw an example of what we term **Positive Distribution Shift**: Even if we have unlimited data from the target test distribution $D_{\text{test}}$, training on a shifted distribution $D_{\text{train}} \neq D_{\text{test}}$ can actually *improve* test performance. This contrasts the typical study of *distribution shift*, i.e. training on one distribution but then applying the predictor, or testing, on another. Typically, it is implicitly assumed that the ideal case would be to train on the test distribution, that training on a different distribution is a compromise, either because we don't know or have access to the true $D_{\text{test}}$, or it's expensive to sample from it, or we have only a limited number of samples and want to supplement them with additional data from related distributions. Distribution shift is usually studied as "how much worse do things get if we train on $D_{\text{train}} \neq D_{\text{test}}$", with answers of the form "if $D_{\text{train}}$ is close or related enough to $D_{\text{test}}$, then it's not much worse". In this paper, we investigate one of several ways in which distribution shift can be *positive*.

Specifically, we systematically study the benefit of such distribution shift when training with mismatched mixing proportions relative to the test distribution. We model the test distribution as a mixture of $K$ components, with known mixing proportions $\{p_k\}_{k=1}^K$, and consider training distributions which are mixtures over the same components but with different mixing proportions $\{q_k\}_{k=1}^K$.

Submitted to 39th Conference on Neural Information Processing Systems (NeurIPS 2025). Do not distribute.

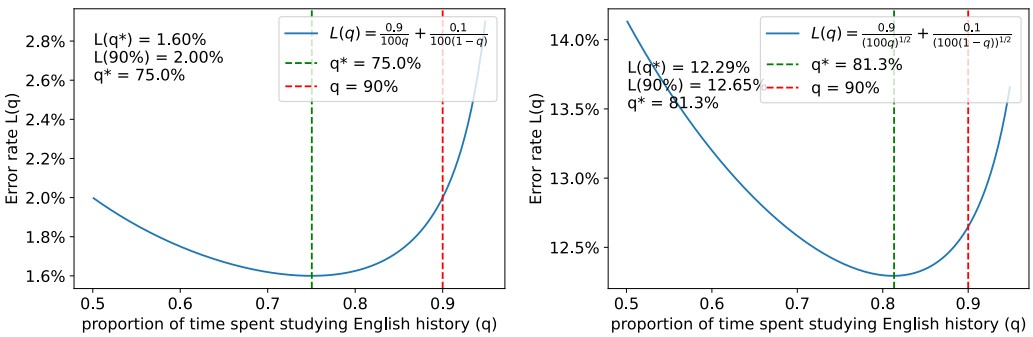

Figure 1: We plot the error rate for a hypothetical scenario modelling the high stakes exam described in Section 1. We model the error rate on each of the test portions as being proportional to $\propto \frac{1}{n_i^\alpha}$, where $n_i$ represents the studying budget spent on that portion of the exam, so $i = 1$ corresponds to European History and $i = 2$ to the Chinese History and set $n_1 + n_2 = N$ to be the total studying budget, with $N = 100$ hours. The exponent $\alpha$ is $\alpha = 1$ on the left plot and $\alpha = 2$ on the right plot. In both cases, we consider $n_1 = qN$ and $n_2 = (1 - q)N$, where $q$ is the proportion of time spent studying for the European History portion of the exam. This way, the error rate on the exam can be written as a function of $q$ as $L(q) = 0.9\frac{1}{(100q)^\alpha} + 0.1\frac{1}{(100q)^\alpha}$. We can see on both plots that shifting away from the testing proportion (red line, i.e. $q = 90\%$) can lead to a better error rate with the optimal test proportion (green line, i.e. $q^*$ whose values are displayed accordingly). See also Corollary 3.3.

We can either think of this as providing guidance when we can actively control mixing between different known components, or as helping us understand how and why a mismatched training distribution can actually be beneficial. In Section 5 we discuss how the analysis is also applicable to a setting where we are not testing on a mixture, but rather on compositional tasks, requiring composing multiple skills, and the skills appear with differing frequencies—this compositional setting served as a major motivation for our study.

We consider different per-component learning curves, capturing different error decays, differing hardness among the components, and the possibility of transfer between components. In Section 3 we consider power law error decay, both the $1/n$ decay mentioned earlier and more general power laws, including with differing component hardnesses or error decays. In Section 4 we consider learning curves corresponding to "fact memorization" scenarios (discussed in Section 4), including those applicable to the skill composition setting, and which correspond to coupon-collector type learning curves. In Section 6 we consider the possibility of transfer between components. In all of these, we show that a mismatched training distribution can be beneficial, characterize the optimal training mixture, and the extent to which mismatch can improve test performance and reduce the training complexity.

Beyond all the specific scenarios, we then argue, in Section 7, that benefiting from mismatch is not the exception but rather the rule. We show that only in rare situations (either measure zero or satisfying a conservation property that does not generally hold) is the optimal training distribution equal to the test distribution, while in "most" cases shift is good.

## 2   Setup

**Learning Setup and Loss**   For concreteness, let $\ell(h, \boldsymbol{z})$ be the loss function that describes how well a model $h$ performs on and instance $\boldsymbol{z} \in \mathcal{Z}$. For example, in supervised learning, $\boldsymbol{z}$ can be an input-output pair $(\boldsymbol{x}, y)$, and $\ell(h, \boldsymbol{z})$ can be the prediction error of $h(\boldsymbol{x})$ vs $y$. Or, in next-word prediction, $\boldsymbol{z}$ can be a document and $\ell(h, \boldsymbol{z})$ can be the average cross-entropy loss when using $h$ to predict each of the next tokens in the document. In any case, for a test distribution $D_{\text{test}}$ over $\boldsymbol{z}$, we evaluate the model through the *test loss* $\mathcal{L}_{D_{\text{test}}}(h) := \mathbb{E}_{\boldsymbol{z} \sim D_{\text{test}}}[\ell(h, z)]$.

**Test Distribution.**   We consider test distributions consisting of a mixture of $K$ components $\mathcal{D}_1, \ldots, \mathcal{D}_K$. A mixture $\mathcal{D}_{\boldsymbol{p}} = \sum_k p_k \mathcal{D}_k$ is then specified by mixing proportions $\boldsymbol{p} =$

$(p_1, \ldots, p_K) \in \Delta_K$ on the probability simplex $\Delta_K$. We let $\boldsymbol{p}$ be the mixing proportions in the test distribution, i.e. $D_{\text{test}} = \mathcal{D}_{\boldsymbol{p}}$, and so the test loss is $\mathcal{L}_{\mathcal{D}_{\boldsymbol{p}}}(h) = \mathcal{L}_{\boldsymbol{p}}(h)$, where here and elsewhere we use the subscript $\boldsymbol{p}$ to denote the mixture $\mathcal{D}_{\boldsymbol{p}}$.

**Learning Algorithm.** We consider abstract "learning algorithm" $\mathcal{A}$, which, given training data (or sequence of training examples) $S \in \mathcal{Z}^N$ of size $N$, outputs a model $\mathcal{A}(S)$ with test loss $\mathcal{D}_{\boldsymbol{p}}(\mathcal{A}(S))$.

**Training Distribution.** We consider training on i.i.d. samples $S \sim \mathcal{D}_{\boldsymbol{q}}^N$ from mixtures $\mathcal{D}_{\boldsymbol{q}}$ of the same $K$ components, but with potentially different mixing proportions $\boldsymbol{q} \in \Delta_K$. For training mixing proportions $\boldsymbol{q}$, we denote $L_N(\boldsymbol{p}, \boldsymbol{q}) = \mathbb{E}_{S \sim \mathcal{D}_{\boldsymbol{p}}^N}[\mathcal{L}_{\boldsymbol{p}}(\mathcal{A}(S))]$ the expected test error on $D_{\text{test}} = \mathcal{D}_{\boldsymbol{p}}$ when training with $D_{\text{train}} = \mathcal{D}_{\boldsymbol{q}}$ (we frequently drop the subscript $N$ if its clear from context). The "non-shifted" expected test loss is then denoted $L_N^{\text{same}}(\boldsymbol{p}) = L_N(\boldsymbol{p}, \boldsymbol{p})$. In contrast, we denote $L_N^*(\boldsymbol{p}) = \min_{\boldsymbol{q} \in \Delta_K} L_N(\boldsymbol{p}, \boldsymbol{q})$ the test error with the best mixing ratios, and $\boldsymbol{q}^*$ the minimizing ratios. When $L^* < L^{\text{same}}$ and so $\boldsymbol{q}^* \neq \boldsymbol{p}$, this means we can benefit from mismatched training. **Our main analysis objective is to charactarize $\boldsymbol{q}^*$, $L^*$ and the improvement over $L^{\text{same}}$.**

We can measure the mismatch benefit through the improvement in test error for a fixed training budget $L_N^{\text{ratio}} = L_N^*/L_N^{\text{same}}$. Or, we can consider the training complexity $N_\epsilon(\boldsymbol{p}, \boldsymbol{q}) = \min N$ s.t. $L_N(\boldsymbol{p}, \boldsymbol{q}) \leq \epsilon$ and the improvement $N_\epsilon^{\text{ratio}} := \frac{N_\epsilon^*(\boldsymbol{p})}{N_\epsilon^{\text{same}}(\boldsymbol{p})}$.

**Specifying the Learning Model** The expected test loss $L_N(\boldsymbol{p}, \boldsymbol{q})$, and so $\boldsymbol{q}^*$ and the benefit of mismatch, depend on the data distributions and learning behaviour of the algorithm. We capture these by modeling the *subpoluation error function* $e_k(\boldsymbol{n})$, i.e. the error on each component $\mathcal{D}_k$ when training with $n_i$ examples from each component $\mathcal{D}_i$. That is, for a vector of sample sizes $\boldsymbol{n} = (n_1, \ldots, n_K) \in \mathbb{Z}_{\geq 0}^K$, denote $\boldsymbol{\mathcal{D}^n} = (\mathcal{D}_1)^{n_1} \times \cdots \times (\mathcal{D}_K)^{n_K}$ the distributions over samples with $n_i$ examples from each component $\mathcal{D}_i$. Then $e_k(\boldsymbol{n}) = \mathbb{E}_{S \sim \boldsymbol{\mathcal{D}^n}}[\mathcal{L}_{\mathcal{D}_k}(\mathcal{A}(S))]$. When $e_k(\boldsymbol{n}) = g_k(n_k)$ depends only on the amount of within-component data, we say the components are *orthogonal*, meaning there is no transfer between them (as in our Chinese and European history example). The scalar function $g_k(n_k)$ then captures the *learning curve* for each component. But more generally, there might also be transfer, with data from one component helping learning on another.

In any case, the learnability function $e : \mathbb{Z}_{\geq 0}^K \to \mathbb{R}^K$, captures our "learning model". In each Section, we consider different forms of learning models and characterize $\boldsymbol{q}^*$ and $L^*$ for these models.

**Data Sets and Training Sequences** In our analysis, we refer to the training budget $N$ and our learning model specifying learning based on $n_k$ examples per component $k$. We can think of $N$ and $\boldsymbol{n}$ as specifying the number of training examples, in which case the training complexity is a sample complexity. Or, we can think of $N$ as indicating the number of training steps, and $n_k$ as indicating the number of steps in which an example from component $k$ is used. In this case, training complexity is a measure of training time. Either interpretation is valid. But we should emphasize that we only study a dependence on *how many* examples are used from each component, *not* on the *order* (as in curriculum learning).

**Learnabilities and Mixing Ratios.** We model learning as a function of the *number* of examples from each component, but for our analysis, it will useful to introduce the function $\bar{e}_{N,k}(\boldsymbol{q}) = \mathbb{E}_{S \sim (\mathcal{D}_{\boldsymbol{q}})^n}[\mathcal{L}_k(\mathcal{A}(S))]$, which captures the expected error on component $k$ with mixing proportions $\boldsymbol{q}$. We will refer to $\bar{e}_k(\boldsymbol{q})$ as the subpopulation error function in terms of the mixture $\boldsymbol{q}$. Since the per-component counts $\boldsymbol{n}$ are multinomial, we have $\bar{e}_N(\boldsymbol{q}) = \mathbb{E}_{\boldsymbol{n} \sim \text{Mult}(\boldsymbol{q}, N)}[e(\boldsymbol{n})] \in \mathbb{R}^K$ and $L_N(\boldsymbol{p}, \boldsymbol{q}) = \langle \boldsymbol{p}, \bar{e}_N(\boldsymbol{q}) \rangle$. Frequently for large sample size $N$, $\bar{e}_N(\boldsymbol{q})$ will concentrate around $e(\boldsymbol{q}N)$, and we will sometimes exploit this in the analysis, or analyze for $\bar{e}(\boldsymbol{q}) \approx e(\boldsymbol{q}N)$.

# 3 Orthogonal Power Law

Many machine learning tasks can be captured with power law error functions. Some classic examples include linear regression or learning VC classes, both of which have error rate $\propto \frac{1}{n}$, where $n$ is the number of data samples. More recently, there have been many papers studying the loss curves for large language models for various tasks as a function of the compute budget in various scaling laws, such as the Chinchilla Scaling Law [Hoffmann et al., 2022].

To model these situations, we will first consider a setup where each of the $K$ tasks is orthogonal and their subpopulation error functions in terms of the number of samples follow a simple power law.

**Model 3.1** (Orthogonal Power Law Error Tasks). There are $K$ orthogonal tasks, each of which takes data from one of the $K$ subpopulations $\mathcal{D}_i$ that appear in the test distribution with probability $p_i$ and whose subpopulation error function $e_k(\boldsymbol{n})$ follows a power law, i.e. $e_k(\boldsymbol{n}) = \frac{A_k}{n_k^{\alpha_k} + B_k}$ for some $A_k > 0, B_k \geq 0$, and $0 < \alpha_k \leq 1$.[1]

In Proposition 3.2, we characterize the test error improvement from the positive distribution shift from optimal data mixing ratios in Model 3.1 when the size of the training data $n$ is large.

**Proposition 3.2** (Optimal Data Mixing Ratios For General Power Law). *In Model 3.1, if for the exponents it holds that $\alpha_1 = \alpha_2 = \cdots = \alpha_S < \alpha_{S+1} \leq \alpha_{S+2} \leq \cdots \leq \alpha_K$ for some $S$ then there exist $\varepsilon_1, \varepsilon_2 \geq 0$ that depend on $\alpha_i$ such that for any test data mixing ratio $\boldsymbol{p}$ and any $n > n_0(A_i, B_i, \alpha_i, p_i)$ we have that the following holds*

$$q_i^* = \frac{1}{N^{\frac{\alpha_i - \alpha_1}{\alpha_i + 1}}} \left( \frac{(\alpha_i p_i A_i)}{\left( \sum_{i=1}^{S} (\alpha_i p_i A_i)^{\frac{1}{\alpha_1 + 1}} \right)^{\alpha_1 + 1}} \right)^{\frac{1}{\alpha_i + 1}} + o\left( \frac{1}{N^{\frac{\alpha_i - \alpha_1}{\alpha_i + 1}}} \right) \tag{1}$$

$$L^{\text{same}}(\boldsymbol{p}) = \frac{1}{N^{\alpha_1}} \sum_{i=1}^{S} p_i^{1-\alpha_1} A_i + o\left( \frac{1}{N^{\alpha_1 + \varepsilon_1}} \right). \tag{2}$$

$$L^*(\boldsymbol{p}) = \frac{1}{N^{\alpha_1}} \left( \sum_{i=1}^{S} (\alpha_i p_i A_i)^{\frac{1}{\alpha_i + 1}} \right)^{\alpha_1} \left( \sum_{i=1}^{S} \frac{(p_i A_i)^{\frac{1}{\alpha_i + 1}}}{\alpha_i^{\frac{\alpha_i}{\alpha_i + 1}}} \right) + o\left( \frac{1}{N^{\alpha_1 + \varepsilon_2}} \right). \tag{3}$$

*The $o(\cdot)$ notation hides dependence on $A_i, B_i, p_i, K$ and $\alpha_i$.*

Proposition 3.2 shows that in the power law Model 3.1, positive distribution shift from optimal data mixing ratios improves the prefactor of the test error dependence on the number of data samples $N$ but does not change the decay rate in terms of $N$. For the proof of Proposition 3.2 and a more precise statement, see Appendix A.1.

To show that this can have significant implications for making training more data efficient, we show the improvement from this positive distribution shift on the sample complexity in the case where we have one majority population and $K - 1$ minority populations that all have the same power exponent $\alpha$. This will also include the test-taking example from Section 1.

**Corollary 3.3** (Sample Complexity Improvement From Optimal Data Mixing For General Power Law). *Consider Model 3.1 with $S = K$, i.e. $\alpha_1 = \cdots = \alpha_K = \alpha$ and $A_1 = \cdots = A_K = A$ with $\boldsymbol{p} = (p, \frac{1-p}{K-1}, \ldots, \frac{1-p}{K-1})$. We have that for any $\epsilon > 0$*

$$N_\epsilon^{ratio}(\boldsymbol{p}) \leq (1-p) + 2\frac{\alpha + 1}{\alpha} \left( \frac{p}{1-p} \right)^{\frac{1}{\alpha + 1}} K^{-\frac{\alpha}{\alpha + 1}}.$$

*Furthermore, the optimal mixing ratios are given by $q_1^* \propto p^{\frac{1}{\alpha + 1}}$ and $q_i^* \propto \left( \frac{1-p}{K-1} \right)^{\frac{1}{\alpha + 1}}$ for $i \geq 2$.*

Corollary 3.3 demonstrates an example case, that if we have one majority population and a number of minority populations, the positive distribution shift from optimal data mixing ratio significantly improves sample complexity. For fixed $p$, if $K$ is large enough, $N^{\text{ratio}}(\boldsymbol{p})$ will be close to $N^{\text{ratio}}(\boldsymbol{p}) \approx 1 - p < 1$, i.e. we get sample complexity improvement of up to $p$. For example, for $p = 0.7$, $\alpha = 0.28$, and $K = 100$, for any $\epsilon > 0$, $N_\epsilon^{\text{ratio}}(\boldsymbol{p}) \approx 0.75$, i.e. we achieve the same error with $\approx 25\%$ less samples. We illustrate this in Figure 2. For the proof of Corollary 3.3, see Appendix A.1.

Furthermore, the test taking example considered in the introduction Section 1 follows from Corollary 3.3, by taking $K = 2$, $\alpha = 1$, and $\boldsymbol{p} = (0.9, 0.1)$. In particular, this shows that the optimal studying budget allocation is $\boldsymbol{q}^* = (0.75, 0.25)$ and the improvement is $N^{\text{ratio}}(\boldsymbol{p}) = 0.8$. This means that if you study for the exam with the right mixing ratio $\boldsymbol{q}^*$, you would need to study $20\%$ less time to achieve the same score as compared to using the test mixing ratio $\boldsymbol{p}$. Further, taking $\alpha = \frac{1}{2}$ we get the second example on Figure 2. This shows that we indeed get $\boldsymbol{q}^* = (0.812\ldots, 0.188\ldots)$ and $N^{\text{ratio}}(\boldsymbol{p}) = 0.944$.

---

[1] We will also use the convention that if $B_k = 0$ then $e_k(\boldsymbol{n}) = \min\{C_k, \frac{A_k}{n_k^{\alpha_k}}\}$ for some $C_k > 0$. This will prevent $L(\boldsymbol{p}, \boldsymbol{q})$ from blowing up to infinity.

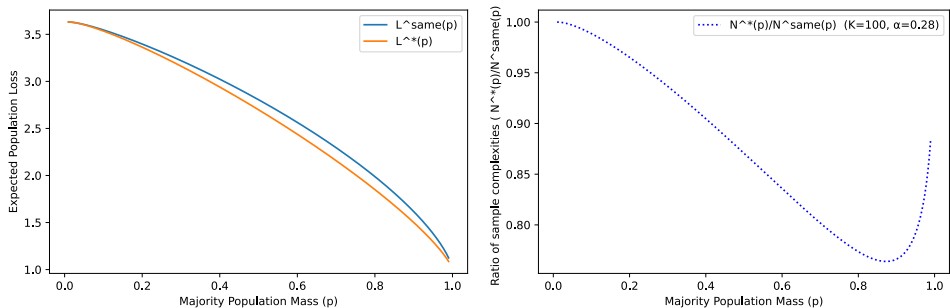

Figure 2: We consider the setup of Corollary 3.3 with $A = 1$, $\alpha = 0.28$, $K = 100$, and some fixed $N$. On the left plot, we show the "non-shifted" expected population loss $L^{\text{same}}(\boldsymbol{p})$ and the optimally mixed expected population loss $L^*(\boldsymbol{p})$ as a function of majority population mass $p$. On the right plot, we show the ratio of sample complexities for any fixed $\epsilon > 0$, $N_\epsilon^{\text{ratio}}(\boldsymbol{p})$ as a function of the mass of the majority population, $p$. We can see significant improvement in the sample complexity from the positive distribution shift from using optimal mixing ratio, even up to $\approx 25\%$.

## 4 Orthogonal Memorization Tasks

We consider a task of memorizing a number of unique elements from a dataset of fixed size, where the test distribution is a mixture of the tasks we are trying to memorize.

**Model 4.1** (Orthogonal Memorization Tasks). Suppose there are $K$ tasks, each of which is a memorization of a unique element. The test distribution is a mixture of these $K$ tasks, where the $k$-th task appears with probability $p_k$. In this case the subpopulation error functions in terms of $\boldsymbol{n}$ is given by $e_k(\boldsymbol{n}) = \mathbf{1}_{\{n_k=0\}}$.

The following theorem characterizes the test error improvement from the positive distribution shift from optimal data mixing ratios in the Orthogonal Memorization Task Model 4.1.

**Theorem 4.2** (Optimal Data Mixing Test Error Improvement For Orthogonal Memorization Task). *In Model 4.1, for all $\boldsymbol{p} \in \Delta^{K-1}$ with $p_1 \geq p_2 \geq \cdots \geq p_K$, the expected loss when training on $n$ samples is given by*

$$L^{\text{same}}(\boldsymbol{p}) = \sum_{k=1}^{K} p_k (1 - p_k)^N \tag{4}$$

$$L^*(\boldsymbol{p}) = (K_N(\boldsymbol{p}) - 1)\delta_N(\boldsymbol{p}) + \sum_{k=K_N(\boldsymbol{p})+1}^{K} p_k, \tag{5}$$

*where $\delta_N(\boldsymbol{p}) \in [p_{K_N(\boldsymbol{p})+1}, \ p_{K_N(\boldsymbol{p})})$ and $K_N(\boldsymbol{p})$ is defined as follows:*

$$K_N(\boldsymbol{p}) := \max\left\{ s \leq K : \sum_{k=1}^{s-1} (1 - (p_s/p_k)^{1/(K-1)}) < 1 \right\}. \tag{6}$$

To understand the magnitude of the test error improvement in Theorem 4.2, we will assume that the test proportions $\boldsymbol{p}$ follow a power law $p_k = \Theta(k^{-\alpha})$ for some $\alpha > 1$ and that the number of tasks to memorize $K$ is larger than the size of the training set $N$. In this case, we show that the improvement from positive distribution shift Theorem 4.2 improves even the test error scaling in terms of $N$. For the proof of Theorem 4.2, see Appendix A.2.

**Corollary 4.3** (Test Error Improvement For Orthogonal Memorization Taks with Power Law Test Mixing Ratios). *If $p_k = \Theta(k^{-\alpha})$ for some $\alpha > 1$ and $K = \Omega(N)$, then*

$$L^{\text{same}}(\boldsymbol{p}) = \Theta(N^{-1+\frac{1}{\alpha}}), \qquad L^*(\boldsymbol{p}) = \Theta(N^{-\alpha+1}).$$

For example, when $\alpha = 1.5$, we have $L^{\text{same}}(\boldsymbol{p}) = \Theta(N^{-1/3})$ and $L^*(\boldsymbol{p}) = \Theta(N^{-1/2})$. For the proof of Corollary 4.3, see Appendix A.2.

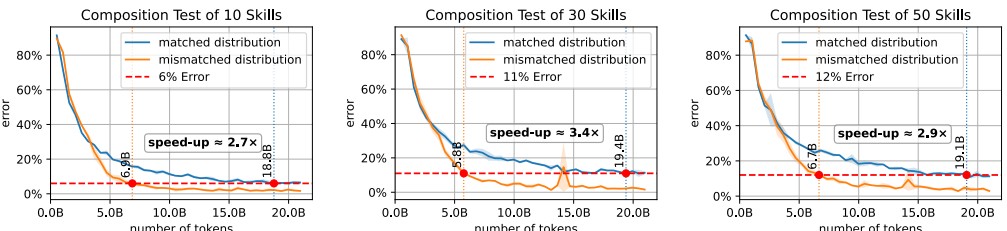

Figure 3: Mismatched distribution improves the test accuracy of a language model in solving a synthetic CoT reasoning task on skill composition (Section 5). During test, the model is asked to compose several functions following a power law. Instead of training directly on this task (blue curve), mixing with another task that uniformly samples the functions improves the final accuracy (orange curve).

## 5 Connection to Skill Composition

All the above analyses focus on the case where tasks are orthogonal. However, if we already know that the test distribution can be decomposed into $K$ tasks, then maybe we should deal with these $K$ tasks independently. So why do we have test mixing ratios in the first place?

We note here that in some cases, we may need to compose these $K$ tasks later at inference time, and the test mixing ratios can come from the proportions in the composition. Imagine that we are training a language model to do mathematical reasoning. Each problem may involve several math skills, and a language model can acquire a math skill only if it sees the skill enough times during training. This can be conceptually modeled as the orthogonal memorization task discussed above, but at inference time, the language model has to sequentially apply the math skills in its chain of thought (CoT). The natural distribution of math skills then determines the test mixing ratios we care about.

We demonstrate this in a concrete synthetic task on skill composition. There are $K$ skills, where the $i$-th skill is a function $g_i$ that maps a number from $\{0, \ldots, 9\}$ to $\{0, \ldots, 9\}$. Each skill has a unique English name. Assume that all these skills are randomly sampled: the names are uniformly random from a name set, and each $g_i$ is uniformly random among all possible functions that map from $\{0, \ldots, 9\}$ to $\{0, \ldots, 9\}$. At inference time, a set of $k$ skills $g_{i_1}, \ldots, g_{i_k}$ are sampled IID following a power law with exponent $\alpha = 1.5$. The language model is prompted with the names of these skills and a number $x \in \{0, \ldots, 9\}$: "`[x] -> [skill name 1] -> [skill name 2] -> ··· -> [skill name k]`". The model is expected to output the result after function composition: $y = g_{i_k}(g_{i_{k-1}}(\cdots g_{i_1}(x)\cdots))$.

Let $D_{\text{test}}$ be the distribution of the above prompt and a CoT calculating the correct answer, with $M = 10^5$, $k$ sampled uniformly from 10 to 50. Is the best strategy just training on the same distribution ($D_{\text{train}} = D_{\text{test}}$)? Inspired by our calculation for the orthogonal memorization task above, properly adjusting the occurrence probability for each skill may lead to better test accruacy. To demonstrate this, we construct another distribution $\mathcal{D}_{\text{uniform}}$ consisting of strings in the form of "`[x] [skill name] = [expected output]`", where the skill and input number are uniformly sampled. In Figure 3, we conduct experiments with a model with GPT-2 architecture and ∼50M parameters. We show that training with $D_{\text{train}} = 30\% \cdot \mathcal{D}_{\text{uniform}} + 70\% \cdot D_{\text{test}}$ significantly outperform training with $D_{\text{test}}$ directly. We defer the experiment details to Appendix C.

## 6 Non-orthogonal Tasks and Transfer Learning

Many transfer learning setups, such as multi-task learning of linear classifiers over linear representation with feature learning Baxter [2011], Maurer [2009], Pontil and Maurer [2013], Aliakbarpour et al. [2024] and multi-task learning with shared sparsity Wang et al. [2016, 2017], the subpopulation error functions $e_k(\boldsymbol{n})$ can be written in the form $e_k(\boldsymbol{n}) = \frac{A_{0,k}}{(n_1 + \cdots + n_k)^{\alpha_k}} + \frac{A_{1,k}}{n_k^{\alpha_k}}$. For example, in multi-task learning of shared sparsity Wang et al. [2017], the error bound takes this form with $\alpha_1 = \cdots = \alpha_K = 1$.

To model all of these cases, we consider the following model of transfer learning.

**Model 6.1** (Standard Transfer Learning Model). There are $K$ subpopulations, each of which appears in the test distribution with proportion $p_k$. The subpopulation error functions depend on the number of samples $\boldsymbol{n}$ as $e_k(\boldsymbol{n}) = \frac{A_{0,k}}{(n_1 + \cdots + n_k)^{\alpha_k}} + \frac{A_{1,k}}{n_k^{\alpha_k}}$, for some $A_{0,k}, A_{1,k} > 0$ and $0 < \alpha_k \leq 1$.

Interestingly, the Standard Transfer Learning Model 6.1 is equivalent to the setup of Orthogonal Power Law Tasks Model 3.1 in the sense that we can understand optimal data mixing ratio $q^*$ and the error improvement of the Standard Transfer Learning model from a specific instance of the Orthogonal Power Law model. Namely, the transfer term in each of the subpopulation loss functions can be decomposed into a transfer error term and a specific task error term $e_k(\boldsymbol{n}) = e_k^{\text{transfer}}(\boldsymbol{n}) + e_k^{\text{spec}}(\boldsymbol{n})$, where $e_k^{\text{transfer}}(\boldsymbol{n}) = \frac{A_{0,k}}{(n_1 + \cdots + n_k)^{\alpha_k}}$ is independent of the distribution of samples across different tasks, and $e_k^{\text{spec}}(\boldsymbol{n}) = \frac{A_{1,k}}{n_k^{\alpha_k}}$ only depends on $n_k$. Therefore, the transfer error term $e_k^{\text{transfer}}(\boldsymbol{n})$ in each of the subpoluation error functions will only offset the final expected loss $L(\boldsymbol{p}, \boldsymbol{q})$ by $\sum_{i=1}^K p_i \frac{A_{0,k}}{N^{\alpha_k}}$, which only depends on the total number of samples $N$. On the other hand, the specific task error terms $e_k^{\text{spec}}(\boldsymbol{n})$ can be thought of as orthogonal tasks and will behave tha same as in Model 3.1. So, for the Standard Transfer Learning Model 6.1, the optimal data mixing ratio $q^*$ and the expected test losses $L^*(\boldsymbol{p})$ and $L^{\text{same}}(\boldsymbol{p})$ are given by Equation (1) and Equation (2) respectively in Proposition 3.2 with $A_k$ being replaced by $A_{1,k}$.

## 6.1 Data Mixing Transfer Learning.

Ye et al. [2025] consider the problem of estimating the outcome performance of a large langue model trained on a mixture of domains. In particular, they find that an exponential function over the linear combinations of mixing proportions leads to good prediction. Namely, they fix the training budget $N$ and only vary the mixing ratio $q$ and show that the validation loss on $i$-th domain can be predicted well by a function of the form $c_i + b_i \exp\left(-\sum_{j=1}^K t_{ij} q_j\right)$, where $c_i, b_i, t_{ij}$ are parameters to fit. Following their work, we propose the following model for the Data Mixing Transfer Learning.

**Model 6.2** (Data Mixing Transfer Learning). There are $K$ subpopulations, each of which appears with probability $p_k$ in the test distribution. Each of the subpopulation error functions in terms of the mixing ratio $q$ are $\bar{e}_k(\boldsymbol{q}) = c_k + b_k \exp\left(-\sum_{j=1}^K t_{ij} q_j\right)$ for some constants $c_k$ and $b_k > 0, t_{ij}$.

We note that even though Model 6.2 is indeed not defined by the subpopulation error functions $e_k(\boldsymbol{n})$, it is precisely the setup that Ye et al. [2025] consider. This slightly deviates from our main setup, which focuses on specifying models by their error functions. However, when the number of samples $N$ is large, it is reasonable to make the approximation that $e_k(\boldsymbol{n}) \approx e_k(\boldsymbol{q}N)$, and Model 6.2 can be interpreted as being defined by the subpopulation error functions of the form $e_k(\boldsymbol{n}) = c_k(|\boldsymbol{n}|) + b_k(|\boldsymbol{n}|) \exp\left(-\sum_{j=1}^K t_{ij}(|\boldsymbol{n}|) n_j\right)$, where $c_k, b_k,$ and $t_{ij}$ are functions that depend only on the total compute budget $N = |\boldsymbol{n}|$.

The following proposition characterizes the test error improvement from the positive distribution shift coming from the optimal data mixing ratio in the data mixing transfer model.

**Proposition 6.3** (Optimal Train Data Mixing Ratio for Data Mixing Transfer Learning Model). *In Model 6.2, if the coefficients $t_{ij}$ are such that $\boldsymbol{T}$ is invertible and and $(\boldsymbol{T}^T)^{-1}\boldsymbol{1} > 0$, and $p_i \neq 0$ for all $i$, the following hold*

$$\boldsymbol{q}^* = (\boldsymbol{T})^{-1}\left(\frac{1 + \boldsymbol{1}^\top \boldsymbol{T}^{-1}\tau}{\boldsymbol{1}\boldsymbol{T}^{-1}\boldsymbol{1}}\boldsymbol{1} - \tau\right)$$

$$L^{\text{same}}(\boldsymbol{p}) = \sum_{i=1}^K c_i p_i + \sum_{i=1}^K p_i b_i \exp\left(-\sum_{j=1}^K t_{ij} p_j\right)$$

$$L^*(\boldsymbol{p}) = \sum_{i=1}^K c_i p_i + \exp\left(\frac{-1 - \boldsymbol{1}^\top \boldsymbol{T}^{-1}\tau}{\boldsymbol{1}^T \boldsymbol{T}^{-1}\boldsymbol{1}}\right)\boldsymbol{1}^T (\boldsymbol{T}^\top)^{-1}\boldsymbol{1},$$

*where $\tau$ is a vector with entreis $\tau_l = \log\left(\frac{[(\boldsymbol{T}^\top)^{-1}\boldsymbol{1}]_l}{p_l b_l}\right)$.*

Proposition 6.3 shows the positive distribution from the optimal data mixing for Model 6.2. Note that the additional conditions on $\boldsymbol{T}, p_i$ are technical conditions used in order to simplify presentation. For the complete statement and the proof of Proposition 6.3, see Appendix A.3.

To demonstrate how large the gap can be, we consider the problem of data mixing transfer learning Model 6.2 with $K = 2$ tasks and a one-directional transfer from the second to the first task.

**Corollary 6.4** (Optimal Data Mixing Ratio Can Have Significant Improvement in the Transfer Learning Model). *Let $K = 2$, let $\boldsymbol{p} = (\frac{1}{2}, \frac{1}{2})$, and let $b_1 = b_2 = b > 0$. If $\boldsymbol{T} = \begin{pmatrix} 1 & \alpha \\ 0 & 1 \end{pmatrix}$ then we have that*

$$L^{\text{same}} - L^* = 2be^{-\frac{1}{2}} \left( 1 - \frac{1}{4}\alpha + O(a^2) \right).$$

*Furthermore, if we let $C = \frac{c_1 + c_2}{2}$ and $B = be^{-\frac{1}{2}}$ then we have that*

$$L^{ratio} = \frac{L_N}{L^*} = \frac{C - B}{C + B} + \frac{BC}{2(B + C)^2}\alpha + O(\alpha^2)$$

Corollary 6.4 shows that for two tasks with a small of transfer between the second to the first we can have error improvement from the positive distribution shift by mismatching training and test distribution, that is $L^{\text{ratio}} \approx \frac{C-B}{C+B} < 1$ for small $\alpha$. For the proof of Corollary 6.4, see Appendix A.3.

# 7 It's Almost Always Better to Mismatch

So far, we have shown the existence of and quantified the positive distribution shift coming from mistmatched test and train data mixing ratios for the cases of orthogonal power law tasks in Section 3, orthogonal memorization tasks in Section 4, and standard transfer learning and data mixing transfer learning in Section 6. that positive distribution shift from mismatching test and train mixing ratios exists. In this section, we will provide further mathematical justification that a positive distribution shift coming from the data mixing ratio almost always exists. That is, we show that it's almost always better to mismatch the training and test distributions: $\boldsymbol{q}^* \neq \boldsymbol{p}$ and $L^*(\boldsymbol{p}, \boldsymbol{q}^*) < L^{\text{same}}(\boldsymbol{p})$.

More precisely, we will show that either the test data mixing ratio is on a measure zero set of the simplex or the subpopulation error functions $e_k(\boldsymbol{n})$ have to be very specific functions, which are meaningless. For example, in the case of orthogonal tasks, either the test mixing ratio is on a measure zero subset or the subpopulation error functions $e_k(\boldsymbol{n})$ are all constants, which we show in Corollary 7.4.

We define the probability simplex $\Delta^{K-1} := \{\boldsymbol{p} \in \mathbb{R}^K : \boldsymbol{p} \geq 0, \ |\boldsymbol{p}| = 1\}$, and its interior $\Delta_+^{K-1} := \{\boldsymbol{p} \in \mathbb{R}^K : \boldsymbol{p} > 0, \ |\boldsymbol{p}| = 1\}$, where $|\boldsymbol{p}| := \sum_{k=1}^{K} p_k$. We will define $f_k(\boldsymbol{p})$ by extending the domain of each $\bar{e}_k(\boldsymbol{p})$ to the set of non-zero, non-negative vectors $\mathbb{R}_{\geq 0}^K \setminus \{\boldsymbol{0}\}$ by defining $f_k(\boldsymbol{p}) := \bar{e}_k(\frac{\boldsymbol{p}}{|\boldsymbol{p}|})$. We further define $L^{\text{same}}(\boldsymbol{p}) := \sum_{k=1}^{K} p_k f_k(\boldsymbol{p})$, which extends the definition of $L^{\text{same}}$ to the set of non-zero, non-negative vectors $\mathbb{R}_{\geq 0}^K \setminus \{\boldsymbol{0}\}$.

**Condition 7.1** (Conservation Condition). $(f_1(\boldsymbol{p}), \ldots, f_K(\boldsymbol{p})) = \nabla L^{\text{same}}(\boldsymbol{p})$ *for all* $\boldsymbol{p} \in \mathbb{R}_{\geq 0}^K \setminus \{\boldsymbol{0}\}$.

**Theorem 7.2** (Positive Distribution Shift Almost Always Exists For Data Mixing). *For any set of subpopulations $\mathcal{D}_1, \ldots, \mathcal{D}_K$ and any learning algorithm $\mathcal{A}$, either Condition 7.1 holds, or there exists a zero-measure set $U$ on $\Delta^{K-1}$ such that for all $\boldsymbol{p} \in \Delta^{K-1} \setminus U$, $L_N^*(\boldsymbol{p}) < L^{\text{same}}(\boldsymbol{p})$.*

Theorem 7.2 shows that either $\boldsymbol{p}$ is on a measure zero set $U$ on $\Delta^{K-1}$ or the Conservation Condition 7.1 must hold. We will show that Conservation Condition 7.1 happens only for very specific cases of subpopulation error functions.

**Conservation Condition Rarely Holds.** First, we will show that if the subtasks are orthogonal, the conservation condition Condition 7.1 is only satisfied if all of the subpopulation error functions are constants.

**Lemma 7.3** (Orthogonal Tasks). *If $K \geq 3$, and if for all $k \in [K]$, $f_k(\boldsymbol{p}) = g_k(\frac{p_k}{|\boldsymbol{p}|})$ for some function $g_k$, then Condition 7.1 holds if and only if $g_k$'s are all constant functions.*

Theorem 7.2 and Lemma 7.3 together show that in the case of orthogonal tasks, positive distirbution shift always exists by changing the training data mixing ratio away from the test mixing ratio, unless all the subpopulation error functions are constant.

**Corollary 7.4** (Positive Distribution Shift Always Exists for Orthogonal Tasks). *For any set of $K \geq 3$ subpopulations $\mathcal{D}_1, \ldots, \mathcal{D}_K$ and any learning algorithm $\mathcal{A}$, if there exists subpopulation $k \in [K]$ such that its error function $e_k$ is not a constant functions over $[N]$ where $N$ is the number of total samples then there exists a measure zero set $U$ on $\Delta^{K-1}$ such that for all $\boldsymbol{p} \in \Delta^{K-1} \setminus U$ positive distribution shift from data mixing exists in the sense that there is $\boldsymbol{q}^* \neq p$ for which $L_N(\boldsymbol{p}, \boldsymbol{q}) = L^*(\boldsymbol{p}) < L^{\text{same}}(\boldsymbol{p})$.*

Further, we show that if the Conservation Condition 7.1 is satisfied, then one function $f_i$ determines the rest up to a constant.

**Lemma 7.5.** *If both $(f_1, \ldots, f_K, L^{\text{same}})$ and $(\hat{f}_1, \ldots, \hat{f}_K, \hat{L}^{\text{same}})$ satisfy Condition 7.1, and if $f_i = \hat{f}_i$ for some $i \in [m]$, then for all $k \neq i$, $f_k(\boldsymbol{p}) = \hat{f}_k(\boldsymbol{p}) + C_k$ for some constant $C_k$.*

The above Lemma 7.5 implies that for every $k$ and corresponding error function $e_k(\boldsymbol{n})$, there exists at most one tuple of error functions $\{e_j\}_{j=1, j\neq k}^{K}$ (up to a individual constant offset for each error function $e_j$) that positive distribution shift does not happen for $\boldsymbol{p}$ of positive measure. This further implies the following corollary.

**Corollary 7.6** (Positive Distribution Shift *Almost* Always Exists for General Tasks). *For any set of $K \geq 3$ subpopulations $\mathcal{D}_1, \ldots, \mathcal{D}_K$ and any learning algorithm $\mathcal{A}$, for all $\boldsymbol{p} \in \Delta_+^{K-1}$, the configuration of $[e_k(\boldsymbol{n})]_{k\in[K], \boldsymbol{n}}$ that positive distribution shift does not happen is zero-measure.*

Corollary 7.6 shows that either the test mixing ratio $\boldsymbol{p}$ is on a set of measure zero on the simplex or the configuration of subpopulation error functions $e_k(\boldsymbol{n})$ is on a set of measure zero. This implies that positive distribution shift exists *almost* always.

# 8 Related Works

**Distribution Shift That is Not Harmful.** The benefits of mismathcing the training and test distribution has already been in studied in some settings. González and Abu-Mostafa [2015] demonstrate in many linear regression problems that mismatched training and test distributions can outperform matched ones. Unlike in our paper, they do not restrict to changing the train distribution only through data mixing, so their results do not fit our framework. On the other hand, we explicitly characterize the positive distribution shift, while González and Abu-Mostafa [2015] only show its existence for linear regression problems and are only able to characterize the distribution explicitly in very special cases. Canatar et al. [2021] show how in high-dimensional kernel regression problems to numerically optimize the training distribution for better test performance. However, they do not characterize the positive distribution shift, but rather only show how to numerically find it for kernel regression. Similarly, they do not restrict the test distribution to one coming from a data mixture, so their results do not fit our framework.

**Data Mixing.** There a number of recent empiricaly works that consider the same setting of data mixing as we do. Ye et al. [2025] introduce data mixing laws, quantitative empirical predictions of large language model performance based on the data mixture proportions. Furthermore, they show experimental results demonstrating that their approach significantly decreases the number of steps needed to reach certain performance. This paper informed our data mixing transfer model and fits in our framework. Goyal et al. [2024] show that data curation for VLMs cannot be compute agnostic. They introduce neural scaling laws that allow for estimating performance on multiple data pools without jointly training on them. Their work fits our framework. Similarly, we also find that optimal mixing ratios are not compute agnostic, specifically in the orthogonal power law tasks, orthogonal memorization task, and standard transfer learning task. Jiang et al. [2025] introduce an algorithm for online optimization of data distributions, that adjusts mixture based on the estimated per-domain learning potential, achieving comparable or better performance than previous methods while maintaing compuatational efficiency. While all of these works consider the same phenomena of changing the training mixing ratio to improve test performacne, the main difference between our work and theirs is that we consider positive distribution shift from data mixing ratio in a broader context and from the theoretical standpoint as well.

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
