# OpenReview forum: "Shift is Good: Mismatched Data Mixing Improves Test Performance"
_NeurIPS.cc/2025/Conference — Submitted to NeurIPS 2025_

### Official Review · Reviewer_8TeD · 2025-06-23

**Clarity:** 1
**Significance:** 2
**Originality:** 2
**Rating:** 3
**Confidence:** 3

**Summary:**

### Summary

This paper challenges the conventional assumption that training and test ratios should be matched for optimal performance. Instead, it demonstrates that mismatching training data ratios is desirable and allows for better generalization performance. The authors provide theoretical support for their claims and show that "positive distribution shift" can provide learning speedups in one GPT2-like model.

**Questions:**

* How would this work translate to frontier models/tasks?
* Why is there a spike in mismatched learning for the middle and right of Figure 3?
* How would other test distribution ratios affect the ideal mismatched ratio? For example, instead of 90:10, it becomes 50:50?
* How many data points are used for these evaluations? From the Appendix it seems like 400 points which can be too small.
* Could the quality of the sampled training points affect these results?

**Ethical Concerns:**

["NO or VERY MINOR ethics concerns only"]

**Final Justification:**

This work presents a valuable idea for controlling the training distribution by first sampling a validation set, which could have potential benefits for pre-training large models, particularly multilingual LLMs. However, a significant weakness is that the paper does not empirically demonstrate this benefit, which I can understand given the limitations in resources. Due to this, I would rate the paper as borderline.

**Limitations:**

Would be good if the authors could include a section for the Limitations of this work, as per declared in the paper checklist.

**Paper Formatting Concerns:**

No issues with formatting.

**Quality:**

2

**Strengths And Weaknesses:**

### Strengths:
* The authors provide theoretical support for the claims made in this paper.
* The insights from this work hold potential for advancing LLM training, particularly for multi-task and multilingual foundation models.


### Weaknesses:
> Major

* While the current experimental setup demonstrates improvements for a training distribution modified from 90:10 to 75:25 when the test distribution is 90:10, its robustness to other test distribution ratios, such as 100:0, remains unclear. It seems intuitive that more training samples in the dominant class (e.g., English) would benefit the model in such scenarios. Furthermore, given that real-world deployment often involves unknown test distribution ratios, it is uncertain how well this approach would generalize.

* This paper lacks explainability and would benefit from deeper analysis on why/where the improvements come from. For example, Fig 1 could show the individual test errors for English and Chinese, as opposed to the current overall error. In a similar vein, the authors could consider metrics that account for imbalances within the data, such as F1 or Recall.

* Given the current limited experiments, I am not convinced that this work would readily translate to larger models like LLaMA (fine-tuning could be considered if pretraining resources are limited). Additionally, since this seems to be a generic paper, I encourage the authors to demonstrate wider applicability to other machine learning tasks, such as computer vision.

> Minor

* Additionally, this paper has too many typos e.g., L77, L83, L328. Figure 1 is also not referenced.

* Quality:
Proofs seem correct, though i did not check rigourously.

* Clarity:
Very poorly written. This paper has too many typos and grammatical errors, leading to poor readability.

* Significance:
Empirical analysis and evaluation on frontier models is lacking.

* Originality:
The theoretical framing of this work appears original.

---

> ### Author Rebuttal · Authors · 2025-07-31
>
> We thank the reviewer for their feedback.
>
> **Q:** robustness to other distributions:
>
> **A:** See answer to question #3.
>
> **Q:** What if the test distributions were uknown:
>
> **A:** The main way we view the paper is providing an answer to the question "why might it be better to train on a different distribution then the test distribution". This is a phenomenon encountered in practice especially with training large models with many proficiencies, with much of model training having to do with selecting good training distributions (for a fixed test distribution). We feel this is a current and important question that is not sufficiently acknowledged or studied, or even made explicit, especially from a theoretical perspective. We do also view our work as providing guidance in this "training distribution selection" problem, in the sense of understand what to look for and what direction or kind of training distribution we want, even in setting where the exact proportions are unknown. Nevertheless, we can also imagine some settings in which the proportions are known, e.g. based on quick analysis of a sample from the test distribution.
>
>
>
> **Q:** explainability:
>
> **A:** We view our results as understanding one purely statistical aspect of positive distribution shift, where the improvement in test error comes from upsampling the components that have a larger impact on the test error and downsampling the components that have a smaller impact on test performance. For example, in the introductory example, because of the power law error scaling, the first few additional training examples in both the Chinese and European histories decrease the error significantly more than a new training data after we have already seen a number of training data in that component. It turns out that 10% of the training data is small enough that an additional example from the Chinese component would decrease the total error more than an additional example from the English component (and of course in expectation). This is the source of positive distribution shift.
>
> **Q:** It is unclear whether our results would hold for large models:
>
> **A:** We believe that our results and their implications apply to even large models such as LLaMA (for pretraining and finetuning) given that empirically power law has been shown to capture the error decay of these models well [1-4]. We agree with the reviewer that further experiments in this direction are welcome and leave that for future work.
>
> Here we provide answers to reviewer’s questions:
>
> **Q1:**
> >How would this work translate to frontier models/tasks?
>
> **A1:** Frontier models are usually used for a variety of tasks - e.g. text editing, code generation, chat, reasoning etc., and furthermore are trained using a variety of different datasets. The loss here would be the pre-training loss, which is empirically demonstrated to follow power law scaling, one of the cases we cover [1]. Furthermore, the phenomena on having improvement from training the model on a different train distribution than a test distribution is encountered in practice with training large models with many proficiencies, with much of model training having to do with selecting good training distributions (for a fixed test distribution). We do also view our work as providing guidance in this "training distribution selection" problem, in the sense of understand what to look for and what direction or kind of training distribution we want.
>
> **Q2:**
> >Why is there a spike in mismatched learning for the middle and right of Figure 3?
>
> **A2:** Loss spikes are common in training LLMs, and most of them are recoverable, i.e., after a few steps, the loss gradually goes back to the value before the spike. Figure 3 shows the average curves of 3 training runs, and we observe a spike in one of them. We believe that adding more runs can eventually average out the spikes and produce a smoother curve.
>
> **Q3:**
> >How would other test distribution ratios affect the ideal mismatched ratio? For example, instead of 90:10, it becomes 50:50?
>
> **A3:** Proposition 3.2 (and Corollary 3.3.) shows the effect of test distributions on the ideal mixing ratio. Taking $K=2$, $A_1=A_2=A$, $\textbf{p}=(p,1-p)$ and $\alpha=1$, we can see that $\textbf{q}^{\star}= \frac{1}{p^{\frac{1}{1+\alpha}}+(1-p)^{\frac{1}{1+\alpha}}}(p^{\frac{1}{1+\alpha}},(1-p)^{\frac{1}{1+\alpha}})$. From this, we see what the improvement is for $p:1-p$ test mixing ratio and what the optimal training distribution $\textbf{p}^{\star}$ is. In this special case, taking $p=1-p=½$, we see that the optimal training distribution is the same as the test distribution $\textbf{p}^{\star} = (½, ½)$ and that for this specific test distribution there is no improvement, i.e. $L^{\star}((½,½))=L^{same}((½,½))$. We note that this is in line with the rest of our theory. Corollary 7.4 says that in any orthogonal task setup with K tasks, as long as one error function is non-constant, then for all $\textbf{p}$ except possibly a zero measure set, there is a positive distribution shift. In particular, for the orthogonal power law setting, if the tasks are equally hard ($A_i$ and $\alpha_i$ are all the same), then the uniform test distribution $\textbf{p} = (\frac{1}{K},\dots, \frac{1}{K})$ will not have a positive distribution shift, i.e. the uniform $\textbf{p}$ is in the measure zero set excluded by Corollary 7.4.
>
> **Q4:**
> >How many data points are used for these evaluations? From the Appendix it seems like 400 points which can be too small.
>
> **A4:** To measure the accuracy of skill composition, we used 400 data points for each evaluation. We argue that 400 data points can already lead to meaningful results because the final test error gap between training with and without mismatch distributions can be as large as 10% in Figure 3.
>
> **Q5:**
> >Could the quality of the sampled training points affect these results?
>
> **A5:** We do online training and we also repeated the experiment 3 times. It seems that the quality of sampled points does not affect the results.
>
>
> [1] Hoffman et al. An empirical analysis of compute-optimal large language model training. NeurIPS 2022.
> [2] Ye et al. Data mixing laws: Optimizing data mixtures by predicting language modeling performance. ICLR 2025.
> [3] Kaplan et al. Scaling Laws for Neural Language Models. 2020.
> [4] Cherti et al. Reproducible scaling laws for contrastive language-image learning. CVPR 2023.

---

> > ### Comment · Reviewer_8TeD · 2025-08-02
> > **Response to Authors**
> >
> > I thank the authors for their response. My central concern regarding the need to know the test distribution beforehand remains unaddressed. This presents a significant challenge for industry practitioners. How should they effectively apply the findings of this work in real-world scenarios? I am also seeking clarification on the rebuttal's mention of a "quick analysis of a sample from the test distribution." It would be helpful to understand what this entails.
> >
> >  I also note that the authors' response to my central concern was identical to the one provided to Reviewer 7UKA. While I appreciate the consistency, a more tailored response would be beneficial. For the rest of my minor concerns, I am satisfied and hope the authors will incorporate them into their paper.

---

> > > ### Author Response · Authors · 2025-08-06
> > > **Reply**
> > >
> > > What we had in mind for “quick analysis of a sample from the test distribution” is, e.g., as follows: if we suspect a mixture structure in the data, we can try to build crude classifiers for the components. This could be very easy, for example, in a language-related task if each mixture component is a different language. Or, in a memorization task, if each mixture component is a topic or area (e.g. sports, science, etc), we can build crude topic classifiers based on a small amount of data for each topic, or perhaps unsupervised clustering of a sample, then classify a sample. Note that the classifier doesn’t have to be very accurate since we don’t care about individual errors, just about the aggregate proportions, and a bit of an error on the proportions is also fine. Using this classifier, we can estimate the unknown mixing proportions.
> > >
> > > We emphasize again that, as we wrote in the response, our main motivation is answering the question “why might it be better to train on a different distribution than the test distribution” and providing high-level guidance, although the above is also a plausible scenario.

---

> > > > ### Comment · Reviewer_8TeD · 2025-08-06
> > > > **Response to Authors**
> > > >
> > > > I thank the authors for their thoughtful and well-written response. The proposed approach of using crude classifiers to estimate mixture proportions appears conceptually similar to standard practices involving hold-out validation sets, where one might derive properties from using a reserved subset.
> > > >
> > > > While this method offers practical utility for preliminary analysis, I would encourage the authors to explicitly clarify this in the paper so as to avoid ambiguity. Can the authors please clarify if my understanding is correct? Thank you.

---

> > > > > ### Author Response · Authors · 2025-08-07
> > > > > **Reply**
> > > > >
> > > > > Thank you for the engaging discussion.
> > > > >
> > > > > Yes, exactly. We can train the classifier on a subset of the training data and estimate the mixing proportions using a separate hold-out set. We will clarify this point in Section 2 (Setup) of the updated version of the paper. Thanks so much for your suggestion!

---

### Official Review · Reviewer_7UKA · 2025-06-30

**Clarity:** 2
**Significance:** 1
**Originality:** 2
**Rating:** 3
**Confidence:** 3

**Summary:**

This paper studies the impact of distribution shift in a multi-task learning setup, where a model is trained and evaluated on data from a mixture of tasks. The authors focus on a specific type of shift: a change in the _proportion_ of tasks between training and testing. They refer to cases where this shift improves performance as “Positive Distribution Shift.” The paper develops theoretical analyses showing that such shifts are often beneficial, and that training on a different task mixture than the test distribution can lead to improved results on this test distribution.

**Questions:**

1. How do you justify the assumption that the task mixture at test time is known?
2. Could your framework and results be extended to cases where the test-time mixture is unknown? How would it affect your results?
3. Could you provide a recap table or summary of theoretical results to help clarify the contributions?
4. Could you discuss how your findings relate to or differ from the MSDA literature (e.g., [B]–[E])?

**Ethical Concerns:**

["NO or VERY MINOR ethics concerns only"]

**Final Justification:**

The paper lacks discussion of the extensive literature on learning under distribution shift, and this remains unaddressed in the rebuttal. Furthermore, I would recommend a better presentation of the practical relevance of the results.

**Limitations:**

The paper should acknowledge that the type of distribution shift studied, i.e. known differences in task proportions, is a special case and differs from more general forms considered in the literature. The paper should also discuss the assumption that the test-time task mixture is known, and consider its implications for real-world applicability.

**Paper Formatting Concerns:**

I did not notice formatting issues in the paper.

**Quality:**

1

**Strengths And Weaknesses:**

### Originality

- **Strengths**: The idea of analyzing task mixture proportions and the link with "data mixture laws" from Ye et al. is interesting and could provide more insights on the optimal combination of dataset in practice.
- **Weaknesses**: The framing of the problem as “distribution shift” is somewhat misleading. The shift considered here (changes in task proportions) differs from how distribution shift is typically defined in the literature. This setup is more closely related to multi-task learning (MTL) and multiple-source domain adaptation (MSDA), which have been extensively studied in prior work ([A]–[E]). The paper does not sufficiently acknowledge or engage with this body of literature.

### Quality

- **Strengths**: The theoretical results are presented across both orthogonal and non-orthogonal tasks settings. An empirical result show how learning on a combination of tasks can improve performance for a single task.
- **Weaknesses**: The assumption that the task mixture at test time is known is strong and not discussed. The findings on optimal task combinations appear to also rely heavily on the assumed power-law form of the error function. Both of these limit the practical applicability of the findings. It would be valuable to explore whether the theoretical insights extend to settings where the test-time task mixture is unknown or must be inferred, which would increase the practical relevance of the findings.

### Clarity

- **Strengths**: The writing is generally clear, and the structure makes it easy to follow the progression of ideas.
- **Weaknesses**: The paper would benefit from a clearer summary of its contributions. A recap table and/or a concluding discussion of the main results would help understand the scope and implications of the work.

### Significance

- **Strengths**: Understanding how task mixtures affect generalization is important for designing robust multi-task learning systems.
- **Weaknesses**: The novelty and significance of the results are somewhat diminished by the lack of engagement with prior work that has studied similar setups. For example, Section 5 and Section 6 revisit how learning on a mixture of multiple tasks can improve transfer to test tasks, which are known findings from MTL and MSDA literature (cf [A] for instance), without clearly distinguishing what is new. Without a clearer differentiation from existing MTL and MSDA literature, it is difficult to assess the novelty of the theoretical findings, which weakens the overall contribution.

### References

[A] Ben-David & Schuller. "Exploiting task relatedness for multiple task learning." Learning Theory and Kernel Machines, COLT/Kernel 2003
[B] Mansour et al. "Domain adaptation with multiple sources." Neurips 2009
[C] Mansour et al. "Multiple source adaptation and the rényi divergence." UAI 2009
[D] Hoffman et al. "Algorithms and theory for multiple-source adaptation." NeurIPS 2018.
[E] Redko et al. "Optimal transport for multi-source domain adaptation under target shift." AISTATS 2019.

---

> ### Author Rebuttal · Authors · 2025-07-31
>
> We thank the reviewer for their feedback.
>
> **Q: Framing the problem as distribution shift:**
>
> **A:** First, multi-source domain adaptation is an instance of distribution shift. Second, regarding the relationship to MTL, we emphasize that the main contribution of our paper is in the case of orthogonal tasks, which is clearly different from the MTL literature. We agree with the reviewer that the non-orthogonal cases are less crisp and the boundary with MTL is less clear. We view this as an additional case of where our framework is applicable and will de-emphasize it in the updated version of our paper.
>
>
> **Q1:**
> >How do you justify the assumption that the task mixture at test time is known?
>
> **A1:** Regarding the assumption that the test mixture distribution is known at test time, the assumption is reasonable if we frame the paper as follows. The main way we view the paper is providing an answer to the question "why might it be better to train on a different distribution then the test distribution". This is a phenomenon encountered in practice especially with training large models with many proficiencies, with much of model training having to do with selecting good training distributions (for a fixed test distribution). We feel this is a current and important question that is not sufficiently acknowledged or studied, or even made explicit, especially from a theoretical perspective. We do also view our work as providing guidance in this "training distribution selection" problem, in the sense of understanding what to look for and what direction or kind of training distribution we want, even in setting where the exact proportions are unknown. Nevertheless, we can also imagine some settings in which the proportions are known, eg based on quick analysis of a sample from the test distribution.
>
> **Q2:**
> >Could your framework and results be extended to cases where the test-time mixture is unknown? How would it affect your results?
>
> **A2:** See the answer above regarding the knowledge of test mixture at test time. We agree with the reviewer that extending the result to the case where the test mixture is unknown is an interesting future direction.
>
> **Q3:**
> >Could you provide a recap table or summary of theoretical results to help clarify the contributions?
>
> **A3:** We systematically study “positive distribution shift” when training with mismatched mixing proportions relative to the test distribution. First, we consider different per-component learning curves and in cases of orthogonal power law, fact memorization, and transfer cases we show that mismatched training and test distributions can be beneficial and we characterize the optimal training mixture and the improvement in test performance and training complexity. Second, we show that only in rare situations the optimal training distribution is equal to the test distribution, while in most cases shift is good. We will add a clearer summary of the main contributions in the updated version of our paper.
>
> | Setup                                                       | Results                                                                                                               |
> |-------------------------------------------------------------|-----------------------------------------------------------------------------------------------------------------------|
> | Orthogonal Power Law Model 3.1                              | Optimal Mixing Ratio and Test Error Improvement Quantified (Proposition 3.2)                                          |
> | Orthogonal Memorization Task Model 4.1                      | Optimal Mixing Ratio and Test Error Improvement Quantified (Theorem 4.2)                                               |
> | Standard Transfer Learning Model 6.1                        | Optimal Mixing Ratio and Test Error Improvement Quantified (Proposition 3.2) (applied to the right parameters as explained in section 6) |
> | Data Mixing Transfer Learning Model 6.2                     | Optimal Mixing Ratio and Test Error Improvement Quantified (Proposition 6.3)                                          |
> | General Orthogonal Case with a Nonconstant Error Function   | PDS always happens (Corollary 7.4)                                                                                     |
> | General Case with Typical Subpopulation error Functions     | PDS always happens (Corollary 7.6)                                                                                     |
>
> **Q4:**
> >Could you discuss how your findings relate to or differ from the MSDA literature (e.g., [B]–[E])?
>
> **A4:** Our work centers on understanding and characterizing how and why might be better to train a different training distribution than the test distribution in a number of relevant settings, and providing guidance in the selection of training distribution problem.
> On the other hand, based on the references [B-D], in the MDSA problem, the learner receives k distributions $D_i$ corresponding to $k$ sources and a predictor $h_i$ for each of the sources (where the predictor could regression, classification, or probability model). The task of the learner is to find a predictor h that performs well on a distribution $D_T$ made up as a mixture of $D_i$ (where the mixture might be either known or unknown to the learner). So, MDSA is an instance of distribution shift.
>
> Our work and the line of work on MDSA differ in a number of ways. The main difference the MDSA literature (including [B-E]) focuses on understanding which predictors $h$ perform well for a given target function and any source distribution $D_T$, while we focus on understanding why and how it might be beneficial to train a model with a different training distribution than a test distribution (and we focus on the case when the test distribution and the error are given by a mixture of components). We also view our work as guiding the training distribution selection problem, in the sense of understanding what to look for and what direction or kind of training distribution we want. None of these is covered by the MDSA literature. In the MDSA literature, there is no notion of a training distribution (or a difference between a training and a test distribution). Namely, in MDSA, the learner is given the domain distributions D_i and the predictors h_i. So the setups (and therefore results) are quite different.

---

> > ### Comment · Reviewer_7UKA · 2025-08-05
> >
> > I would like to thank the authors for their rebuttal and discussing my remarks. I think the addition of the table improves the overall clarity of the results presented.
> >
> > However, I still disagree with the overall framing of the paper around "distribution shift". The authors did not respond to this point. While I agree that "MSDA is an instance of distribution shift", the setting considered in this paper is essentially the same as MSDA and should also be viewed as just one specific instance. My main concern is that the paper introduces a notion of tasks, each with its own distribution, and the shift occurs only in how these tasks are combined between training and testing. In contrast, the general case of distribution shift typically involves a single training task and a single testing task, with no notion of task combination.
> >
> > Regarding Q1, the issue remains that the test distribution is unknown in practice. Therefore, the paper does not truly address the challenge of selecting the best training distribution. As a result, I find the question of "why might it be better to train on a different distribution than the test distribution" less relevant in practice. Since the test distribution is unknown and not accessible, we will always train on a different distribution in this setting.
> >
> > Finally, concerning Q4, when tasks are orthogonal, which is the main setup in the paper, the objective of finding the best mixture of training distributions is similar to constructing predictors for each source and then determining the optimal way to combine them for the test distribution. As reviewer EzzX pointed out, "the properties of models, training algorithms, etc., are all hidden in the power law definition of the error function," making this equivalent to combining predictors that follow similar power laws. Moreover, as mentioned above, the results rely on explicitly knowing the test task mixture, which limits their practical utility for the problem of "guiding the training distribution selection".

---

> > > ### Author Response · Authors · 2025-08-07
> > > **Reply**
> > >
> > > Thank you for your further comments and for an engaging discussion.
> > >
> > > **Q: framing the paper as distribution shift**
> > >
> > > Regarding the framing of the paper as positive distribution shift, as we describe in the paper on lines 21-31, we view this as just one instance of positive distribution shift (and hence distribution shift). In general, we term something a positive distribution shift if training on a shifted distribution $D_{train}\neq D_{test}$ improves performance. Our setup does not solely focus on a combination of tasks (except for the skill composition part). Different components can, e.g., represent different domains of the same task (such as different domains of knowledge in question answering task, or different languages in a language related task).
> > >
> > >
> > >
> > > > I find the question of "why might it be better to train on a different distribution than the test distribution" less relevant in practice.
> > >
> > > Regarding our question "why might it be better to train on a different distribution than the test distribution": we disagree with the view that the question is irrelevant if the component structure of the test distribution is unknown.  In general in ML, we consider test distributions that are not “known” (in the sense that we do not have a description of them), but that we can sample, i.e. collect training examples, from them (this is formalized, eg, as the basis of Vapnik’s and Valiant’s models of statistical/PAC learning). This is often the situation in practice. And even if the test distribution is unknown and samplable, we can choose to collect samples from a different distribution available for us (e.g. if we want to build a hand writting recognizer for 10-year-old handwriting, we can collect data from adult writers, or we can collect data in which we ask writers to write text with atypical letter frequencies). We, in fact, view this as an extremely relevant question since much of what is done these days in training large models is exactly trying to decide which distributions/data sources to collect data from and use in training.
> > >
> > > **Q: power law**
> > >
> > > Regarding the details being hidden in the power law: for orthogonal tasks, the power laws are not that complicated and don’t hide much.  Even with ubiquitous $\frac{1}{m}$ power laws (as in, e.g., linear regression and PAC guarantees) and orthogonal tasks, we get interesting, and as far as we are aware, previously undescribed behaviour.

---

> ### Comment · Reviewer_7UKA · 2025-08-08
>
> Thank you for the additional discussion.
>
> > Our setup does not solely focus on a combination of tasks
>
> The general setup described in Section 2 explicitly considers a test distribution composed of a mixture of tasks. Moreover, the tasks used in the mixture are the same during both training and testing, which reinforces the point that the setup is indeed centered around a fixed combination of tasks.
>
> > Power law
>
> To clarify, my original comment was not about the use of power laws, but about how it is connected to prior work that uses learned predictors with specific error rate.
>
> One of my main concerns, as stated in my initial review, is the lack of engagement in the paper with the extensive literature on learning under distribution shift, and this remains unaddressed in both the rebuttal and the discussions.
> For instance, the whole domain adaptation literature (some references were already provided in my initial review) is directly relevant but never mentioned.
>
> > Relevance in practice
>
> I agree that studying learning under mismatched training and test distributions is practically important. However, that is not the setup studied in the paper. The results provides guidance on finding the best mixture of training tasks given a mixture of the same tasks for testing.
> If the true test distribution (e.g., 10-year-old writers) differs from the approximate one used for evaluation (e.g., adult writers), or even differ from the one(s) available for training, then the result becomes less informative.
>
> I am maintaining my original score. I encourage the authors to include a more thorough discussion of previous work and a better presentation of the practical relevance of the setup studied.

---

### Official Review · Reviewer_EzzX · 2025-07-01

**Clarity:** 3
**Significance:** 2
**Originality:** 3
**Rating:** 3
**Confidence:** 3

**Summary:**

This paper consider a senario where the training and test data distributions can be seen as mixtures of some sub-distribution components. It shows that in some cases, having mismatched training and test distribution, i.e. having different mixing weights in the training and test distributions, may help improve the generalization performance.

**Questions:**

1. Some parts of the writing are too casual. For example, in Line 8-14, the description of the exam example is written too conversationally for an academic paper. Personally, I think this is a good example as a motivation of the work and a warmup for the readers, but this conversational style is more appropriate to be used in oral presentations rather than in a paper.
2. What are the key differences between mixture setting and compositional setting?
3. Line 63. Test distribution is defined as a mixture of some "components". What are the definitions of the components? Are they all distribution but with non-overlapping supports? Or can their supports overlap? Can all distributions be formulated, in some sense, as mixtures of some underlying components (with careful definitions of the components)?
4. Line 72. In the definition of $L_N({p},q)$, from my understanding the expectation should be taken with respect to $S\sim\mathcal{D}^N_q$ not $S\sim\mathcal{D}^N_p$, is that correct?
5. From my understanding, in the setup of this paper, the properties of models, training algorithms, etc. are all hidden in the power law definition of the error function given in Model 3.1, is this correct? If so, is there any justification of choosing the orthogonal power law particularly to formulate the error function? What makes it a better option than (possibly) other choices? I think this should be clearly explained since it seems that the power lar assumption is highly relied on by the main results and their proofs.
6. Lin 107, at the end of this line, I think it should be $\bar{e}_N(q)$, is this correct? If not, then what is the definition of $\bar{e}(q)$?
7. Is there any proof of the claim in Line 106 that "$\bar{e}_N(q)$ will concentrate around $e(qN)$"? Also, is $\bar{e}_N(q)$ a random variable? If so, where does the randomness come from?
8. Proposition 3.2. What are the intuitions of the assumtions? Also, what is the definition of $n_0(A_i,B_i,\alpha_i,p_i)$?
9. Line 79. Does $L^\text{ratio}_N$ represent the improvement or the training budget?
10. Line 86. What is $g$ function? I don't see it defined anywhere, or maybe I have missed it.
11. Corollary 3.3. What is the reason that $A$ and $B_k$'s are not in the bound? Do they have no effect on the sample complexity improvment? If so, then how does the assumption that $A_1=\cdots=A_K=A$ affect the result?
12. Section 4. I am not familiar with this field of works. Can you provide a more concrete introduction of orthogonal memorization tasks? For example, what could be the data? What is the mathematical formulation of a task? What are the objectives of the tasks, etc.? Thank you.
13. Line 240. The concentration property needs to be justified, but doesn't have to be mathematically detailed.
14. Line 275. What's the difference between $\Delta^{K-1}$ defined here and $\Delta_K$ mentioned in the previous text sections.
15. Condition 7.1. What is the intuition of this condition? What does it tell us about the model and/or the test distribution?
16. The related works can be put into a early section, and the last section should include a brief conclusion of the work and addressing of the limitations.

**Ethical Concerns:**

["NO or VERY MINOR ethics concerns only"]

**Final Justification:**

I will keep my original rating. While the presentation and the writing have been perfected, I think the paper still lacks significance and impact to the field. It doesn't provide guidance for practical applications, nor provide interpretations and extensions of prior works.

**Limitations:**

Addressed.

**Quality:**

2

**Strengths And Weaknesses:**

Strenghths:
1. The main result that mismatching can actually be helpful is thought-provoking.
2. The chain of logics is clear.
3. Comprehensive review of related works.

Weaknesses:
1. Some parts of the writing are too casual.
2. Some notations are not well used.
3. Some definitions are not clear.
4. A few results and assumptions are not fully justified.
5. Lack of conclusions and justification of limitations.

---

> ### Author Rebuttal · Authors · 2025-07-31
>
> We thank the reviewer for their feedback. We provide answers to the reviewer’s questions.
>
> **Q1:**
> >Some parts of the writing are too casual. For example, in Line 8-14, the description of the exam example is written too conversationally for an academic paper. Personally, I think this is a good example as a motivation of the work and a warmup for the readers, but this conversational style is more appropriate to be used in oral presentations rather than in a paper.…
>
> **A1:** We appreciate the reviewer’s feedback regarding the writing style. We feel that such a conversational style is useful for the introduction. We note that the conversational style is kept only in the introduction - we make everything formal and precise in the technical sections. If something is not precise enough, we would be very happy to make it more precise.
>
> **Q2:**
> >What are the key differences between mixture setting and compositional setting?
>
> **A2:**
> Our compositional setting can be viewed as a mixture setting as follows: we show that mixing the original data distribution $D_{test}$ with an auxiliary distribution $D_{uniform}$ can lead to a lower test error. In the notation of Section 2, we have $D_1 = D_{test}, p_1 = 1$ and $D_2 = D_{uniform}, p_2 = 0$.
> We emphasize the compositional setting in Section 5 because it is conceptually related to the orthogonal memorization setting in Section 4. In particular, the compositional setting can be understood as learning $N$ orthogonal skills and composing them together at inference time. However, unlike the orthogonal memorization, it is not possible to separate the test distribution in the compositional setting directly into a series of independent components.
>
> **Q3:**
> >Line 63. What are the definitions of the components? Are they all distribution but with non-overlapping supports? Or can their supports overlap? Can all distributions be formulated, in some sense, as mixtures of some underlying components (with careful definitions of the components)?
>
> **A3:** The supports of the distributions do not have to be disjoint. For example, in the Europan and Chinese history case, a data sample could be relevant to both Chinese and European history, so it should be counted as a sample for both components. Not all distributions can be represented in this way, but this condition still captures many behaviors - see answer to question 5.
>
> **Q4:**
> >Line 72. In the definition of L_N(p,q), from my understanding, the expectation should be taken with respect to S~D_q^N and not S~D_p^N, is that correct?
>
> A4: Yes, this is a typo. The expectation should be taken with respect to $S\sim D^{N}_{q}$.
>
> **Q5:**
> >From my understanding, in the setup of this paper, the properties of models, training algorithms, etc. are all hidden in the power law definition of the error function given in Model 3.1, is this correct? If so, is there any justification of choosing the orthogonal power law particularly to formulate the error function? What makes it a better option than (possibly) other choices? I think this should be clearly explained since it seems that the power lar assumption is highly relied on by the main results and their proofs.
>
> **A5:** Yes, in the setup of this paper, the learning model is specified by the subpopulation error functions $e_k(\textbf{n})$. We study a few cases of different error functions capturing different problems - the orthogonal power law, the orthogonal memorization task (where the error functions follow a 0-1 law), and the transfer case. Why orthogonal power law is an excellent question. First, we take the tasks to be orthogonal as that is the base case when there is no transfer. Secondly, we take the power law because it captures many behaviors based on theoretical analysis (e.g. realizable PAC, agnostic PAC, linear regression) and also empirically ([1], [2], [3], [4]).
>
> **Q6:**
> >Line 107, at the end of this line, I think it should be \bar{e}_N(q), is this correct? If not, then what is the definition of \bar{e}_N(q)?
>
> **A6:** Yes, at the end of Line 107 it should have been $\bar{e}_N(\textbf{q})$, and in the sentence before on Line 106 it was supposed to say $e(\textbf{n}), \textbf{n}\sim \text{Mult}(\textbf{q},N)$ will concentrate around $e(\textbf{q}N)$. The proof for the concentration property in the orthogonal case is in Appendix A1, in particular Propositions A1-A4.
>
> **Q7:**
> >Is there any proof of the claim in Line 106 that "\bar{e}_N(q) will concentrate around e(qN)"? Also, is \bar{e}(qN) a random variable? If so, where does the randomness come from?
>
> **A7:** Line 106 was supposed to say that “$e(\textbf{n})$ for $\textbf{n} \sim \text{Mult}(\textbf{q},N)$ will concentrate around $e(\textbf{q}N)$ and we will sometimes exploit this in the analysis, or analyze for $\bar{e}_N(\textbf{q}) \approx e(\textbf{q}N)$”. The statement was not meant as a formal general claim but rather an indication of what the outline of the main argument is. We thank the reviewer for pointing out these typos.
>
> **Q8:**
> >Proposition 3.2. What are the intuitions of the assumptions? Also, what is the definition of n_0(A_i,B_i,\alpha_i,p_i)?
>
> **A8:** Note that rewriting $\alpha$ in this way does not decrease generality. The intuition is that the error decay will be dominated by the component with the slowest error decay (i.e. smallest $\alpha$) so we want to write the main terms of the error using these components. For the ease of presentation, we didn’t write the definition of the n_0(A_i,B_i,\alpha_i,p_i) because the formula is messy and long but it can be found in the appendix in Equations 9-12 and 14-15.
>
> **Q9:**
> >Line 79. Does L_N^{ratio} represent the improvement or the training budget?
>
> **A9:** $L_N^{\text{ratio}}$ represents the error improvement with the same training budget. $N_{\epsilon}^{\text{ratio}}$ represents the improvement in the training budget to achieve an error at most $\epsilon$.
>
> **Q10:**
> >Line 86. What is g function?
>
> **A10:** They are defined on line 89, i.e. they are any scalar functions. We will clarify that sentence in the updated version of the paper.
>
> **Q11:**
> >Corollary 3.3. What is the reason that A and B_k's are not in the bound? Do they have no effect on the sample complexity improvement? If so, then how does the assumption that A_1=..=A_k=A affect the result?
>
> **A11:** In this special case when all $A_i$ are equal, the sample complexity improvement is independent of $A$. This can be seen from the fact that the subpopulation error functions and hence the loss will multiplicatively depend on $A$, so the ratio will not depend on $A$. Corollary 3.3 deals with the special case when all tasks are equally hard (e.g. the introductory example). The dependence can be seen from Proposition 3.2. and Equations 2 and 3. If not all $A_i$ are the same it would introduce an extra term of the form $\frac{\left( \sum_{i=1}^{S} (\alpha_i p_i A_i)^{\frac{1}{1+\alpha_i}}\right)^{\alpha_1}\left( \sum_{i=1}^{S} \frac{(p_i A_i)^{\frac{1}{\alpha_1}}}{\alpha_i^{\frac{\alpha_i}{\alpha_i+1}}}\right)}{\sum_{i=1}^{S}p_i^{1-\alpha_i}A_i}$.
>
> **Q12:**
> >Section 4. I am not familiar with this field of works. Can you provide a more concrete introduction of orthogonal memorization tasks? For example, what could be the data? What is the mathematical formulation of a task? What are the objectives of the tasks, etc.?
>
> **A12:** Let $S$ be the set of possible elements to memorize and $s_1,\dots, s_k \in S$. Assume that the learning rule memorizes all elements it has seen so far and let $M$ be the set of elements the model has seen. Let task $i$ be memorizing element $s_i$, i.e. we get error $0$ if $s_i\in M$ and error $1$ if $s_i \notin M$. For example, this could model fact retrieval or question answering using LLMs. The data in this case would be the text used for (pre)training the LLMs.
>
> **Q13:**
> >Line 240. The concentration property needs to be justified, but doesn't have to be mathematically detailed.
>
> **A13:** The approximation is reasonable because $e_k(\textbf{n})$ has a nice behaving closed form and $\textbf{n}$ concentrates around $\textbf{q}N$. The proof would be similar to the orthogonal case (Proposition A2). We will add a short justification to the updated version.
>
> **Q14:**
> >Line 275. What's the difference between \Delta^{K-1}  defined here and \Delta_K mentioned in the previous text sections.
>
> **A14:** Yes, they are the same thing. We will update section 7 to make the notation consistent.
>
> **Q15:**
> >Condition 7.1. What is the intuition of this condition? What does it tell us about the model and/or the test distribution?
>
> **A15:** Condition 7.1 represents the very special case when there is no positive distribution shift, so the intuition is that it restricts the subpopulation error function to a very specific set of functions that do not represent a typical case. In the orthogonal case, Condition 7.1 implies that all the subpopulation error functions in terms of the mixture are constant. Lemma 7.5 further shows that Condition 7.1 uniquely determines the subpopulation error functions in terms of the mixture up to a constant.
>
> **Q16:**
> >The related works can be put into an early section, and the last section should include a brief conclusion of the work and addressing of the limitations.
>
> **A16:** We will add a conclusion section at the end and also address limitations and future work. Regarding the related works section, we put it at the end of the paper so that a) we get to the main results faster and b) the flow of the paper is more natural this way because our paper is not directly based on any one previous work.
>
> [1] Hoffman et al. An empirical analysis of compute-optimal large language model training. NeurIPS 2022.
> [2] Ye et al. Data mixing laws: Optimizing data mixtures by predicting language modeling performance. ICLR 2025.
> [3] Kaplan et al. Scaling Laws for Neural Language Models. 2020.
> [4] Cherti et al. Reproducible scaling laws for contrastive language-image learning. CVPR 2023.

---

### Official Review · Reviewer_CMS3 · 2025-07-03

**Clarity:** 3
**Significance:** 3
**Originality:** 3
**Rating:** 5
**Confidence:** 4

**Summary:**

The paper investigates a very interesting idea: should the training distribution task mixture ratios match the test distribution rations, with an interesting conclusion, that the answer is no. They show that under power law error decay, often observed in practice, models benefit from slight undersampling of majority task. Even if the tasks are orthogonal to each other.

**Questions:**

It's not a question per se, but more of a future work suggestion, that it would be neat to see how this can be adapted if we don't know the task (or mixture) distribution at test time. Could taking the test samples without labels still build an ideal training set for them, based on this theory, by say approximating similarity of two points by their foundation model embedding similarity (a bit similar to the citation above). The assumption that we know the test-time mixture is a bit limiting in practice now.

**Ethical Concerns:**

["NO or VERY MINOR ethics concerns only"]

**Final Justification:**

I looked over other reviews and I still think this is a very nice paper.

Some of the other reviewers seem to take issue with the assumption that we know the test-time data mixture, but in some fields that's actually quite common. E.g. in drug discovery it's very common that you train on some generic data and know what subset of it you would be testing on. For example you have data for binders to a lot of protein targets, but you know you will only be evaluating your model against some protein targets or similarly you will only use it on some sub-types of proteins, while training distribution might be all types of proteins.

**Limitations:**

yes

**Quality:**

3

**Strengths And Weaknesses:**

I think this a super interesting and important research topic. In practice data and its precise mixture can have tremendous impact on model performance. This paper provides a quite unintuitive and thus very valuable insight, that when mixing different tasks in training data ideally one should slightly undersample the majority tasks, even if the training data exactly matches test-time ratios.

The theory provided is sound, covers both the orthogonal task scenario and transfer learning. It is also backed by appropriate in-silico experiment. While one could desire more extensive experimental validation, I think it is sufficient for a theoretical paper.

The Fig 3 experiments that could be improved if 1) the 30% mixing ratio of uniform and iid weights would be motivated and 2) performance of 100% uniform sampling would be added, to show both extremes (now we only have 100% iid baseline). For example https://arxiv.org/pdf/2410.05980 argue that uniform sampling improves model generalization. So 100% uniform is also an interesting baseline to have.

---

> ### Author Rebuttal · Authors · 2025-07-31
>
> We would like to thank the reviewer for the thoughtful and constructive review. We are grateful that the reviewer found our research topic “super interesting and important,” and our findings offer “quite unintuitive and thus very valuable insight.” We also appreciate your positive comments on the soundness of our theoretical approach and the backing of appropriate experiments. We agree with the reviewer that an interesting future direction is when the task distribution is not known at test time, and we will leave it for future work.
>
> > The Fig 3 experiments that could be improved if 1) the 30% mixing ratio of uniform and iid weights would be motivated and 2) performance of 100% uniform sampling would be added, to show both extremes (now we only have 100% iid baseline).
>
> **A:** Thank the reviewer for the constructive suggestions.
> The mixing ratio 30% here is a hyperparameter, and in our preliminary experiments, we found that **setting this hyperparameter to a number between 10% ~ 70% generally improves the test accuracy**. This method is motivated by our calculation for the orthogonal memorization task: properly upsampling low-probability skills can lead to better memorization (Line 197), and mixing the original data distribution with $D_{uniform}$ is one way to effectively upsample low-probability skills. We will expand the discussion of the motivation in our camera-ready version.
>
> **Using 100% uniform sampling would lead to very high test errors.** It is worth noting that each data point in $D_{uniform}$ only contains one skill (which corresponds to just one step of the CoT for skill composition), and thus the model will fail to learn how to compose skills with CoT if it only trains on $D_{uniform}$. Only by mixing the original data distribution with $D_{uniform}$ can the model learn the skills and how to compose them properly. This is why we mix them together rather than using just one data distribution.
>
> Overall, our experiments here demonstrate that mixing the original data distribution with another distribution can lead to a better test error than just training on the original one. The mixed distribution with 30% mixing ratio we tested is one way to achieve this goal, and we do not claim that the mixed distribution is optimal.

---

### Official Review · Reviewer_EFmf · 2025-07-07

**Clarity:** 2
**Significance:** 1
**Originality:** 1
**Rating:** 2
**Confidence:** 4

**Summary:**

The paper considers cases where one might train and test on distributions with different mixture proportions, arguing that this may be beneficial. This seems like a big claim but actually it’s a bit obvious. Here the mixture components correspond to completely distinct populations and the problem can be such that there is no transfer among them. Thus because error tends to go down with sqrt(n) and the authors are considering the case where the sample budget is fixed and thus we must allocate the finite budget among the mixture components, we will nearly always choose a slightly different weighting of the components than the test example. The result is presented as a counterintuitive finding of “positive distribution shift” but I think this is really just rediscovering simple motivational cases for active learning. Credit to the authors for making these arguments formal and attempting to weave together a nice analysis. However, I think the problems here sit at the level that the conclusions are on their face obvious given the assumptions the authors are willing to make and they recapitulate known facts and existing intuition about active learning.

**Questions:**

None

**Ethical Concerns:**

["NO or VERY MINOR ethics concerns only"]

**Limitations:**

None.

**Paper Formatting Concerns:**

None.

**Quality:**

1

**Strengths And Weaknesses:**

Per above: “The result is presented as a counterintuitive finding of “positive distribution shift” but I think this is really just rediscovering simple motivational cases for active learning.”

---

> ### Author Rebuttal · Authors · 2025-07-31
>
> We thank the reviewer for their feedback.
>
> **Q: This is just rediscovering motivation for active learning:**
>
> **A:** The active learning setting, as typically studied, is rather different, even if relaxed, from the setup of our paper. For example, pool based active learning, where individual examples can be selected from a sample, or query based are both different, and almost always the benefit comes from being adaptive to the data so far.
>
> In any case, we could not find a discussion of a mixture setting with analysis of the optimal proportions and discussion of being better with a train distribution different from the test. If we are missing something, we would be happy to get the references and see what the reviewer has in mind.
>
> **Q: The conclusions are on their face obvious:**
>
> **A:** Furthermore, based on a small informal survey of the experts in the field with whom we discussed our results while writing the paper, the results did not seem so immediately obvious. Yes, the results might be obvious in hindsight because the math is very simple. But we need to ask the reviewer whether the answer to the motivating example in the introduction was immediately obvious to them before reading on.

---

### Decision · Program_Chairs · 2025-09-17

**Decision:**

Reject

**Comment:**

The paper studies a situation where a mismatch between train and test distributions is helpful. In particular, when the data is a mixture of multiple distributions, then under some assumptions it makes sense to downweight the components with higher mixing weight. Assuming a fixed budget of training points and disjoint domains for distributions, it would makes sense to increase the weight of low weight components (so to have data points from all components in your training data). The authors also discuss the optimal weight of setting the weights assuming access to the underlying weight of the clusters.

Overall, while the reviewers appreciated this observation, they were concerned about the framing of the problem as distribution shift, and at the significance of the results. The authors are encouraged to relate the question to learning from imbalanced data, multiple source domain adaptation, and active learning (because there is an overall budget). Also adding the discussion of handling unknown mixing weights would improve the paper.